# A Smartphone-Enabled Continuous Flow Digital Droplet LAMP Platform for High Throughput and Inexpensive Quantitative Detection of Nucleic Acid Targets

**DOI:** 10.3390/s23198310

**Published:** 2023-10-08

**Authors:** Elijah Ditchendorf, Isteaque Ahmed, Joseph Sepate, Aashish Priye

**Affiliations:** 1Department of Chemical and Environmental Engineering, University of Cincinnati, Cincinnati, OH 45221, USAahmedie@mail.uc.edu (I.A.);; 2Digital Futures, University of Cincinnati, Cincinnati, OH 45221, USA

**Keywords:** droplet microfluidics, loop-mediated isothermal amplification, smartphone diagnostics, 3D printed microfluidics, smartphone image analysis

## Abstract

Molecular tests for infectious diseases and genetic anomalies, which account for significant global morbidity and mortality, are central to nucleic acid analysis. In this study, we present a digital droplet LAMP (ddLAMP) platform that offers a cost-effective and portable solution for such assays. Our approach integrates disposable 3D-printed droplet generator chips with a consumer smartphone equipped with a custom image analysis application for conducting ddLAMP assays, thereby eliminating the necessity for expensive and complicated photolithographic techniques, optical microscopes, or flow cytometers. Our 3D printing technique for microfluidic chips facilitates rapid chip fabrication in under 2 h, without the complications of photolithography or chip bonding. The platform’s heating mechanism incorporates low-powered miniature heating blocks with dual resistive cartridges, ensuring rapid and accurate temperature modulation in a compact form. Instrumentation is further simplified by integrating miniaturized magnification and fluorescence optics with a smartphone camera. The fluorescence quantification benefits from our previously established RGB to CIE-xyY transformation, enhancing signal dynamic range. Performance assessment of our ddLAMP system revealed a limit of detection at 10 copies/μL, spanning a dynamic range up to 10^4^ copies/μL. Notably, experimentally determined values of the fraction of positive droplets for varying DNA concentrations aligned with the anticipated exponential trend per Poisson statistics. Our holistic ddLAMP platform, inclusive of chip production, heating, and smartphone-based droplet evaluation, provides a refined method compatible with standard laboratory environments, alleviating the challenges of traditional photolithographic methods and intricate droplet microfluidics expertise.

## 1. Introduction

Testing for infectious diseases and genetic anomalies inherently relies on molecular tests based on nucleic acid analysis. These diseases remain a significant global health burden, accounting for millions of deaths annually [1,2]. Despite advancements in medical science, diseases such as malaria, tuberculosis, HIV/AIDS, and, more recently, COVID-19 continue challenging healthcare systems, particularly in low-resource settings. The rapid transmission of these diseases, combined with factors such as antimicrobial resistance and the emergence of new pathogens, underscores the urgent need for reliable, rapid, and inexpensive molecular testing.

The polymerase chain reaction (PCR) is presently recognized as the benchmark method for molecular tests. PCR amplifies a specific nucleic acid (DNA/RNA) sequence, allowing for the detection of even minute quantities of genetic material. However, the complexities associated with PCR instrumentation, including precise temperature cycling and the need for specialized and expensive equipment, make it suboptimal for use at the point of care. To bridge this gap, there is an increasing demand for point-of-care tests that match their benchtop counterparts’ sensitivity, speed, and accuracy but in a more accessible, cost-effective, and straightforward format. Microfluidic technology has emerged as a promising solution, offering several advantages, such as easy integration, reduced reagent consumption, and the potential for miniaturization. Some microfluidic-based platforms, such as GeneXpert, Iquum LIAT analyzer, and Filmarray, have already been commercialized. However, their high costs (primarily due to their reliance on expensive thermal cyclers and fluorometers) often render them unsuitable for widespread adoption.

To circumvent these challenges, several isothermal nucleic acid amplification methods have been developed, such as strand displacement amplification (SDA), helicase-dependent amplification (HDA), recombinase polymerase amplification (RPA), and loop-mediated isothermal amplification (LAMP) [3,4,5,6,7]. Among these, loop-mediated isothermal amplification (LAMP) stands out due to its rapid results, simplicity, and ability to amplify DNA at a constant temperature. Furthermore, LAMP is often equated with PCR in terms of sensitivity, yet it exhibits a notable resistance to inhibitory substances present in blood and various clinical specimens, leading to simplified sample preparation [8,9,10]. LAMP assays are also integrable with paper-based reaction formats, making them suitable for lateral flow systems [11,12,13]. However, LAMP assays are semi-quantitative, often yielding binary (yes/no) results rather than providing a detailed quantitative analysis of the genetic material present in the sample. It has been noticed that real-time LAMP possesses a limited dynamic range, and target concentrations below approximately 1000 copies/μL lack correlation with the threshold time [14,15]. Additionally, LAMP encompasses a more intricate biochemistry compared to PCR, necessitating 4–6 primers that identify 6-8 unique regions within the target DNA sequence. This complexity predisposes LAMP to non-specific amplification, frequently leading to erroneous positive results [16].

Recently, digital LAMP was introduced to address some of these challenges where the LAMP reaction volume is compartmentalized into discrete micron-sized droplets or compartments (~20–800 μm in size), enabling the parallelization of a LAMP reaction into an equivalent of 10^2^–10^6^ individual reactions. Relative to LAMP, digital LAMP achieves absolute quantification in a single assay without the need for external calibration curves, circumventing variable LAMP efficiencies across tests and resulting in a limit of detection down to a single-molecule level. Current digital LAMP methods can be categorized into either (i) chamber-based setups that utilize microfluidic chambers to compartmentalize LAMP mixture [17,18,19,20,21] or (ii) droplet digital LAMP (ddLAMP) setups that unitize droplet microfluidics to emulsify the LAMP mixture. The chamber-based dLAMP method consolidates compartmentalization of fluidic volume, heating, and optical analysis within a single device, eliminating external transfers and reducing mixture loss. However, the reliance of ddLAMP systems on complex and expensive peripheral components compromises its portability and applicability in the point-of-care setting.

On the other hand, digital droplet LAMP (ddLAMP) employs droplet microfluidics (typically with a flow-focusing structure) to compartmentalize the LAMP mixture [22,23,24,25,26,27,28]. By adjusting the flow rates of oils and LAMP mixtures, the number of droplets can be tailored, typically resulting in increased LAMP compartments and enhancing detection sensitivity and dynamic range. However, the reliance of ddLAMP systems on complex and expensive imaging systems such as a fluorescence microscope or flow cytometer compromises its portability and applicability in the point-of-care setting.

While there have been recent efforts to improve the capabilities of existing ddLAMP systems [28,29,30,31], there is still scope for improvement. First, traditional chamber or flow-based ddLAMP systems involve complex fabrication processes such as multilayer soft lithography and chemical etching which are time and labor-intensive. Second and more importantly, most of the ddLAMP systems rely on sophisticated imaging systems such as fluorescence microscopes or flow cytometers which has mostly restricted their more widespread use. Recently, there have been efforts to perform ddLAMP image analysis using smartphones [27,32] but these systems limit their analysis to chamber-based ddLAMP systems with limited throughput. Here we introduce a smartphone-enabled ddLAMP system that can analyse continuously flowing droplets generated with a 3D-printed microfluidic chip enabling an inexpensive and portable ddLAMP platform. Our platform (i) eliminates the need for tedious photolithographic techniques using disposable 3D-printed droplet generator chips and (ii) eliminates expensive optical microscopes or flow cytometers by harnessing a consumer smartphone with an onboard custom image analysis application. We then characterize the performance of ddLAMP in our platform by quantifying the fraction of positive droplets as a function of target DNA concentration in the sample. Finally, we invoke Poisson statistics to enable absolute quantification of DNA concentration by co-relating the fraction of positive droplets to the initial number of copies of the target DNA molecule in the sample.

## 2. Methods

### 2.1. Chip Fabrication and Microfluidic Operations

We employed our previously developed direct 3D printing approach to create our droplet generator microfluidic chip [33]. The 3D model of the droplet generator chip was made in FreeCAD. Special care was taken to position the microfluidic channels between 1–4 slice layers beneath the chip surface, which mitigated the risk of curing any liquid resin trapped within the channels during the UV exposure process. The channel cross-section was specifically fixed at 200 × 200 microns for the droplet generator design, especially in the flow-focusing section, to ensure optimal droplet generation. The CAD file was then sliced in Prusa slicer software (version 2.3) and uploaded to the SLA printer (Prusa SLS). After printing, we employed compressed air to effectively expel any uncured liquid resin trapped inside the microfluidic channels through the designated inlet and outlet ports. Once air-dried, the chips were subjected to a final curing phase in the Prusa curing station (CW1S) for one minute. This created a functional droplet generator microfluidic chip, with the entire fabrication process spanning approximately 2 h. To authenticate the dimensions of these microfluidic features, we used a brightfield optical microscope (Leica fluorescence microscope; magnification range of 10–100×) to capture high-resolution images. The ToupView software (Ver. 3.7), a specialized microscope image analysis tool, facilitated the precise quantification of the defined length scales and other microfluidic attributes pertinent to droplet generation. The syringe pumps employed in this study are New Era NE-102X models, which operate on a 12 V DC power supply and draw a current of 1000 mA. In the current configuration, the syringe pumps operate on a DC power supply sourced from a wall outlet. However, it is feasible to replace this with a 12 V battery pack to enhance the system’s portability.

### 2.2. Heater Design

Our engineered heating system comprises miniature heating blocks made from machined aluminum. Within these blocks, we embedded two heating cartridges and utilized an NPN transistor to control the heat generated (BD139). The operation of this transistor is driven by a pulse-width modulation (PWM) signal generated by a digital pin on an Arduino microcontroller. We integrated three K-type thermocouples throughout the aluminum block to maintain a consistent temperature within the system. The choice of K-type thermocouples was influenced by their widespread use, cost-effectiveness, and compatibility with open-source microcontrollers such as Arduino. Their reliability and accuracy in temperature measurement (accuracy of ±0.75% and operational temperature range of −200 °C to ~1200 °C) make them ideal for such applications. The output from these thermocouples was directed to a MAX 6675 thermocouple amplifier, which then transmitted the average internal temperature readings to the Arduino microcontroller. In response to this data, the Arduino adjusted the current supplied to the cartridge heater to maintain a consistent temperature within the aluminum block. We incorporated a Proportional-Integral-Derivative (PID) controller for fine-tuned temperature control within the Arduino setup. This controller dynamically modulated the power sent to the cartridge heater. The tuning of this PID controller was done using the Ziegler–Nichols method. By setting the integral and derivative gains to zero and incrementally increasing the proportional gain, we identified the critical gain (K_c_) and the oscillation period (P_c_). Based on these measurements, we determined the optimal PID values to be K_p_ = 0.68, K_i_ = 1.49, and K_d_ = 0.06.

### 2.3. LAMP Reactions

The LAMP master mix formulation included FIP and BIP primers at concentrations of 1.6 μM each, F3 and B3 primers at 0.2 μM each, and both LF and LB primers at 0.4 μM each. The list of primers used to amplify the λ phage DNA target is provided in Appendix A Additionally, the LAMP mix comprised of Bst DNA polymerase (New England Biolabs, Ipswich, MA, USA) at 8000 U/mL, 10× isothermal amplification buffer containing 2 mM of MgSO_4_ (New England Biolabs), an added 6 mM of MgSO_4_, dNTPs at 1.4 mM each, and the SYTO 9 dye at 50 μM. 25 µL of this formulated mix was utilized as the dispersed phase for the droplet microfluidic operations. For assays requiring positive controls, the master mix was supplemented with 1 μL of lambda DNA at specified concentrations. Conversely, the negative control was established by adding 1 μL of distilled water to the master mix.

### 2.4. Smartphone Detection

The methodology for capturing and analyzing droplet images in the digital droplet loop-mediated isothermal amplification (ddLAMP) platform involves several key components and steps, each carefully designed to optimize performance and reliability. The imaging system consists of a smartphone camera that is coupled with a commercially available miniaturized optical microscope attachment, specifically the Carson MicroBrite Plus 60×–120× Pocket Microscope. This configuration yields a magnification range from 60× to 120×, corresponding to a resolution capability that enables the differentiation of features ranging between 10 and 1000 μm in size. Illumination is provided by a 3W RGB LED, serving as the primary excitation light source. This LED is coupled with a blue band-pass filter with an excitation wavelength of 490 nm (Thorlabs MDF-TOM, Newton, NJ, USA). The emitted fluorescence is captured through a strategically aligned 520 nm emission filter (Thorlabs MDF-TOM), which is positioned in line with the smartphone camera. Video capture is conducted at a high-definition resolution of 1080 × 1920 pixels and a frame rate of 60 frames per second (fps). Consistency in imaging is maintained by fixing the ISO and exposure settings at ISO 400 and an exposure duration of 1/60 s, respectively. After capturing, the video frames are separated into their constituent RGB channels and uploaded to a MATLAB online (https://www.mathworks.com/products/matlab-mobile.html, accessed on 2 October 2023) drive via MATLAB mobile running on an Android 13 operating system. During the preprocessing stage, each channel undergoes Gaussian noise reduction. A kernel of dimensions 5 × 5 pixels is used, and the noise reduction algorithm considers a radius of 2 pixels around each target pixel. Gaussian weighting is employed, with a standard deviation of 1.5. Following noise reduction, a median image is computed to serve as a stable background, which is then subtracted from each individual frame to highlight the droplets. The Hough transformation is utilized for robust circle detection in the images. The algorithm is configured with a 1-degree angular resolution and a 1-pixel distance resolution. To refine droplet contours, an erosion operation is performed using a circular structuring element of a 2-pixel radius, followed by a dilation operation with a similar structuring element. Droplet dimensions are converted from pixel units to micrometric measurements using a predetermined conversion factor of 0.5 μm/pixel. For fluorescence quantification, the images undergo a transformation from RGB to CIE-xyY color space. This involves an initial gamma correction (γ = 2.2), followed by a Direct Linear Model transformation to obtain device-independent CIE XYZ tristimulus values. The normalized tristimulus values yield two key metrics: luminance and chromaticity, with luminance serving as a critical parameter for quantifying fluorescence intensity. The final stage involves a thresholding technique, where a threshold value of twice the negative signals is applied to differentiate between positive and non-positive droplets based on their intensity. The image analysis code is provided (Appendix A).

## 3. Results

### 3.1. Smartphone ddLAMP Platform

Our smartphone-based ddLAMP platform integrates two syringe pumps, a 3D-printed microfluidic droplet generator chip, low-powered isothermal heaters, and a smartphone droplet imaging system (Figure 1). The syringe pumps drive the continuous phase (oil) and the dispersed phase, consisting of the LAMP assay mix, into a custom-made 3D-printed microfluidic droplet generator chip. Notably, the chip’s fabrication uses a desktop SLA 3D printer, eliminating the need for conventional photolithography techniques or a clean room environment. The chip is designed with a flow-focusing junction, facilitating the formation of droplets that subsequently navigate through a serpentine channel. For optimal heat transfer, the chip is placed in direct contact with a spring-loaded low-powered heater, which incubates the flowing droplets to a temperature conducive for LAMP, maintained for a residence time of 30 min, which corresponds to the droplets’ transit time through the chip’s serpentine channels. The LAMP mixture is pre-enriched with the fluorescent marker SYTO-9, which exhibits fluorescence upon binding with DNA amplicons.

Consequently, in a positive ddLAMP assay, a subset of droplets emits pronounced fluorescence, termed as positive droplets, while the remainder display dim fluorescence, or negative droplets, attributable to the Poisson distribution of the target DNA within the droplets. Post-heating, the droplets transition to the optical detection system. A smartphone camera interfaced with a microscope, LED excitation source, and fluorescence filters continuously analyzes each droplet to quantify its brightness. The amplification results of the ddLAMP assay are discerned using a custom image analysis software operational on the smartphone. The comprehensive ddLAMP operational protocol, encompassing chip fabrication, heating, and smartphone-centric droplet analysis, offers a streamlined approach that can be seamlessly integrated into most laboratory setups, which obviates the need for time-consuming photolithographic techniques and intricate expertise in droplet microfluidics. Furthermore, to mitigate the risk of cross-contamination, all components, including the 3D-printed chips, tubing, and droplet container, are designed for single use and discarded after each ddLAMP test.

### 3.2. Droplet Generator

The essential phase in digital droplet nucleic acid amplification is the generation of monodispersed droplets. Two prevalent methodologies to achieve this are through T-junctions that involve a perpendicular intersection where one fluid meets another and flow focusing that employs the principle of symmetrically shearing the dispersed phase using the continuous phase [34]. The latter has displayed superiority in consistently generating droplets of uniform size. Adding a narrow section in the focusing section results in the highest shear at the nozzle’s slimmest region, which ensures that droplets consistently separate from the fluid stream, producing uniform droplets. However, the primary technique for fabricating such droplet-generation microfluidic chips relies on photolithography, which is expensive and time-consuming and requires the researcher to have training in cleanroom fabrication facilities, restricting the application of ddLAMP to well-equipped laboratories.

To circumvent this, we utilize our previously developed method for 3D printing microfluidic chips that enables one to fabricate a complete microfluidic chip using a single CAD file in less than 2 h without needing photolithography or tedious chip bonding steps [33]. Briefly, the fabrication technique uses a custom resin formulation with a desktop stereolithography (SLA) based 3D printer to fabricate microfluidic chips reliably and rapidly with embedded channels. The inlets and outlets are interfaced with nylon tubing to facilitate liquid flow with external syringe pumps. With this technique, we fabricated a microfluidic chip (Figure 2A,B) with a flow-focusing droplet generation segment using oil (as the continuous phase) and LAMP assay mix (as the dispersed phase). Once formed, the continuous phase propelled aqueous LAMP assay droplets through 15 serpentine channels (Appendix A). This layout ensured that each droplet’s residence time could be modulated between 10–100 min, contingent on the continuous phase’s flow rate. Our findings indicate that, at a set dispersed flow rate (*Q_D_* = 5 μL/min), manipulating the continuous phase’s flow (*Q_C_*) rate permits control over droplet size (Figure 2C). By increasing the continuous phase’s flow rates, droplet sizes can be reduced. Extremely low values of *Q_c_* lead to droplets reaching their maximal potential size, constrained by channel dimensions, at approximately 200 microns. Conversely, higher *Qc* values result in droplets with smaller diameters, approximately 100 microns. Impressively, the variation in droplet size was observed to be less than ±5 microns across all droplets synthesized using our chip.

### 3.3. Thermal Management

To actuate LAMP assays, droplets need to be incubated at 65 °C for ~15 to 30 min. Conventional ddLAMP systems predominantly utilize a standard thermal cycler or Peltier heating elements for this incubation step. These methods, while effective, may not offer the most integrated and efficient approach for standalone platforms. We engineered low-powered miniature heating blocks comprising two resistive cartridges that generate heat within the machined aluminum blocks (Figure 3A,B). The power delivered to the cartridge heater, which essentially determines the heat output, is regulated by an NPN transistor. This transistor’s operation is governed by a pulse-width modulation (PWM) signal emanating from a digital pin of an Arduino. This setup ensures precise control over the current flowing through the cartridge heater, ensuring accurate heat generation. Three K-type thermocouples were strategically positioned across its length to continuously monitor and regulate the aluminum block’s temperature (Figure 3A,B). These thermocouples relayed the average internal temperature of the heater to the Arduino to provide active feedback to maintain the temperature at the desired setpoint. The Arduino dynamically adjusts the current to the cartridge heater to maintain the desired temperature using a PID controller. The tuning of this PID controller was achieved using the Ziegler–Nichols method. We observed consistent oscillations in the control system by setting the integral and derivative gains to zero and incrementally increasing the proportional gain. A circuit diagram of the Arduino-controlled heating module is provided in Figure 3C.

Our primary objective was to assess the efficacy of our engineered heating chamber in maintaining steady-state temperature operations, a critical requirement for the LAMP reaction. By implementing a Proportional-Integral-Derivative (PID) controller, we successfully regulated the current passed through the cartridge. This regulation ensured that our heater reached the predetermined setpoint in under a minute and maintained a stringent temperature regulation, deviating by no more than ±0.8 °C around the 65 °C setpoint (Figure 3D). Compared to other heating systems used to actuate LAMP, our system offers rapid heating and precise temperature control in a miniaturized and low-powered form factor.

### 3.4. Smartphone-Based Image Analysis

To visualize and quantify the amplification results from LAMP assays, a fluorescence dye (calcein) or a DNA intercalating dye (SYTO-9) is typically added. The ddLAMP droplets incorporating a positive template strand produce a nucleic acid amplification reaction, emitting strong fluorescence signals. The fluorescence signal is then quantified in individual droplets using instruments such as fluorescence microscopes [35], flow cytometers [36], or even customized high-throughput fluorescence detectors [22], which are expensive and not adaptable for point-of-care settings. Here, we eliminate the expensive and bulky optical instrumentation by integrating miniaturized magnification optics and fluorescence emission/excitation optics with a consumer smartphone camera running an onboard image analysis program to quantify ddLAMP products.

The droplets exit the microfluidic chip through the chip outlet and into a thin and transparent PTFE tubing (ID: 300 microns) passed through the optical module attached to the smartphone. The optical magnification with the smartphone camera is achieved by interfacing it with an off-the-shelf miniaturized optical microscope attachment (Appendix A). The microscope attachment offers a magnification range of 60×–120× (resulting in high resolution to discern features in the range of 10–1000 μm), which ensures detailed imaging of the microfluidic tubing and individual droplets. A 3D-printed enclosure houses and aligns the microscope with the smartphone camera while integrating a custom RGB LED interfaced with a blue band-pass filter that serves as the excitation source to illuminate the droplets (Appendix A). Positive droplets emit a green fluorescence signal, filtered and captured by an emission band-pass filter aligned with the smartphone camera.

The image analysis workflow flowchart is presented in Figure 4 and starts with a video capture of continuously flowing droplets with the camera ISO and exposure kept constant. A recording duration of 30 min is chosen to ensure comprehensive capture of all droplets in the microfluidic flow. Post-recording, the image analysis is conducted using MATLAB Mobile, which facilitates extracting individual frames from the video. Once extracted, these image files, representing the RGB channels, are uploaded directly to the MATLAB online drive for automated processing. In the preprocessing phase, each image color channel undergoes Gaussian noise reduction by spatially averaging pixels based on proximity and intensity similarity, effectively reducing any random noise. Subsequently, a median image of all frames is computed and subtracted from each frame. This background subtraction technique emphasizes the droplets against a consistent and uniform background. For droplet detection, the Hough transformation—a robust technique for circle detection—is employed, which identifies the circular shapes of droplets by transforming the image space into a parameter space. Following detection, morphological operations, specifically erosion, and dilation, refine the boundaries of the droplets. Each detected droplet is stored in an object array containing RGB pixel values. This organized array is subsequently passed to dedicated functions for each droplet’s size and fluorescence intensity measurements. The size of the droplets is determined through a pixel-to-micrometer conversion corresponding to the largest chord of the droplet geometry. For fluorescence measurement, we utilize our previously developed RGB to CIE-xyY transformation to enhance the quantification of the captured fluorescent signals This transformation process separates the color and intensity of RGB pixels to produce luminance values with a higher dynamic range [37,38,39,40,41]. Briefly, a gamma transformation is applied to the RGB values, correcting for the non-linear response of the sensor’s pixel intensity values. These corrected values are then processed through the Direct Linear Model, converting them into device-independent CIE XYZ tristimulus values normalized to yield each RGB pixel’s luminance and chromaticity values. The derived luminance discrimination (L) metric quantifies the fluorescence signal’s strength, independent of color. Lastly, in the droplet categorization phase, a thresholding technique is employed. This method determines the luminance cutoff that differentiates between a positive droplet and a non-positive one, ensuring accurate and reliable results in the analysis of droplets in the microfluidic chip.

### 3.5. ddLAMP Quantification

In the process of discerning between positive and negative droplet signals, it is imperative to establish a definitive threshold. For our study, this threshold was ascertained using a fixed value derived from the Luminance values observed in a no-template control test devoid of the DNA target (Figure 5A). Once this threshold was instituted, it facilitated the computation of the proportion of positive compartments. To test the performance of our integrated system for ddLAMP detection, we prepared five distinct samples, each containing lambda phage DNA target at varying concentrations ranging from a final LAMP assay concentration of 1 copy/μL to 10^4^ copies/μL. Additionally, a negative control devoid of DNA served as no template control. These LAMP assay samples were then channeled into the microfluidic droplet generator chip, with oil serving as the continuous phase, facilitating the generation of monodisperse ddLAMP droplets size 100 microns. The synthesized droplets were transported into the heating zone of the microfluidic chip consisting of serpentine channels that equilibrate and maintain each droplet’s temperature to 65 °C. The flow rate was calibrated to ensure a residence time of 30 min for each droplet within the heating zone such that LAMP reactions were actuated within droplets containing DNA targets. Post LAMP reactions, the droplets were transported out of the chip into thin and clear PTFE tubing that passed through the smartphone imaging system, where the luminance values for each droplet were recorded. For better visualization, droplets registering luminance values surpassing the defined threshold were color-coded green, symbolizing positive outcomes.

In contrast, droplets with diminished luminance, indicative of negative results, were rendered in red. A discernible clustering pattern emerged in the luminance space (Figure 5). Positive and negative droplets appeared to congregate around distinct luminance values, with a conspicuous gap separating the two clusters. As hypothesized, the droplets from the negative control exhibited dim fluorescence, corroborated by the diminished luminance values, thereby signaling a negative result (Figure 5A). Conversely, the positive samples between 10^1^–10^4^ copies/μL produced several droplets exhibiting pronounced luminance values, indicative of anticipated positive outcomes (Figure 5C–E). As the DNA concentration increased, there was a corresponding rise in the proportion of positive droplets. This trend is due to the higher chance of droplets containing the target DNA at elevated concentrations, increasing the fraction of positive droplets. The sample with 1 copy/μL did not result in positive droplets (Figure 5B), suggesting the limit of detection of the proposed system is 10 copies/μL, which aligns with other ddLAMP systems [18,23].

By counting all the positive droplets (i.e., droplets with luminance values > threshold luminance), we can determine the fraction of positive droplets (*f_p_*) corresponding to DNA concentration in the sample. As the DNA concentration in the sample is increased from 10 to 10,000 copies/μL, the calculated *f_p_* ranges from 0.005 to 0.95, suggesting a dynamic range of 10^4^ copies/μL (Figure 6A). Additionally, the dynamic range can potentially be increased by increasing the number of droplets and decreasing the volume of droplets.

For absolute quantification, one can use Poisson statistics to correlate the fraction of positive droplets (*f_p_*) in a ddLAMP assay to the initial concentration of target DNA in the sample (*C_o_*). To do this, we first note that the probability that a given droplet contains *k* copies of target molecules can be determined by applying the Poisson probability distribution function [42].
(1)$$P\left(k\right)=\frac{\lambda^{k}e^{-\lambda }}{k!}$$
where *λ* is the average number of target molecules in a droplet, i.e., *C_o_* = *λ*/*V_d_* (*V_d_* is the volume of a droplet).

Then, the mathematical probability that a droplet contains at least one target molecule [1 − *P*(0)] can be experimentally determined by calculating the fraction of positive droplets *f_p_* = *N_P_*/*N_T_* (where *N_T_* and *N_P_* represent the total number of droplets and the number of droplets that are positive respectively).
(2)$$\left[1-P\left(0\right)\right]=\left[1-\frac{\lambda^{0}e^{-\lambda }}{0!}\right]=\left[1-e^{-\lambda }\right]=f_{p}$$

Using the relation between *λ* and *C_o_*, we can re-arrange Equation (2) to determine the initial target concentration from the number of positive droplets counted.
(3)$$C_{o}=-\frac{ln\left[1-\frac{N_{P}}{N_{T}}\right]}{V_{d}}$$

We find that the experimentally calculated values of *f_p_* for varying initial concentrations of DNA (*C_o_*) follow the exponential curve as predicted by Equation (3) (Figure 6B). The divergence from expected values at elevated concentrations can be attributed to the limitations of Poisson statistics in such contexts. While the Poisson distribution presumes each event is independent, the actual probability of a droplet containing a target DNA molecule is influenced by the presence of DNA in preceding droplets.
Figure 6Absolute quantification of target DNA using Poisson statistics. (**A**). Determination of the fraction of positive droplets (*f_p_*) against increasing DNA concentration in the sample, ranging from 10 to 10,000 copies/μL. *f_p_* was calculated by determining *N_P_* (number of positive droplets)/*N_T_* (number of total droplets). (**B**) Experimentally derived *fp* values for varying initial DNA concentrations (*C_o_*) align with the exponential Poisson statistics curve predicted by Equation (3).
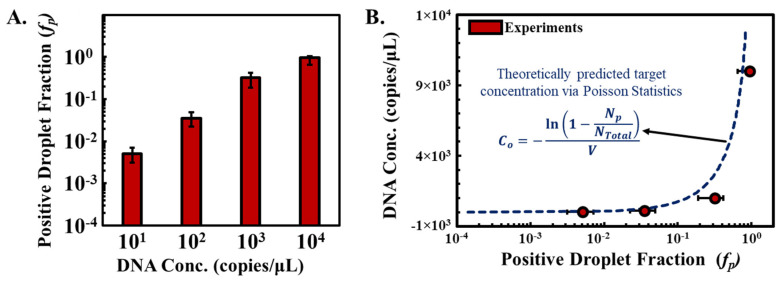


## 4. Conclusions

The recent advancements in droplet-based digital LAMP (ddLAMP) systems have demonstrated significant potential in simplifying and democratizing DNA quantification techniques. In this study, we developed a smartphone-based continuous flow droplet ddLAMP system that integrates a 3D printed microfluidic droplet generation chip with smartphone image analysis to enable quantitative nucleic acid detection. The system effectively generated monodisperse ddLAMP droplets, which, upon undergoing LAMP reactions, emitted fluorescence signals that were captured and characterized by a smartphone camera through a dedicated image analysis program. With concentrations ranging from 1 copy/μL to 10^4^ copies/μL, the ddLAMP’s limit of detection was established at 10 copies/μL with a dynamic range of 10^4^ copies/μL. Additionally, the relationship between the number of positive droplets and initial DNA concentration was delineated using Poisson statistics, with the results largely in agreement with theoretical predictions. The platform offers a suitable alternative to commercially available expensive digital droplet nucleic acid amplification systems (costing upwards of 50,000 USD) at a fraction of the cost, enabling affordable scalability. This scalability brings ddLAMP closer to practical, real-world applications and holds promise for its application in point-of-care diagnostics.

Despite the robustness of the presented system, future iterations of this system could benefit from better statistical models that account for the limitation of Poisson statistics when performing absolute quantification of test samples. Additionally, while smartphone optics presents a cost-effective and portable alternative to traditional fluorescence detectors, calibration across different smartphone models and optimization for various ambient conditions can further increase its adaptability and accuracy. By offering a low-cost, compact, and efficient solution for nucleic acid detection, this ddLAMP system could impact fields such as environmental monitoring, food safety, and personalized medicine. As demonstrated here, the integration of smartphones in scientific instrumentation paves the way for democratized access to advanced diagnostic tools, making them accessible even in resource-limited settings.

## Figures and Tables

**Figure 1 sensors-23-08310-f001:**
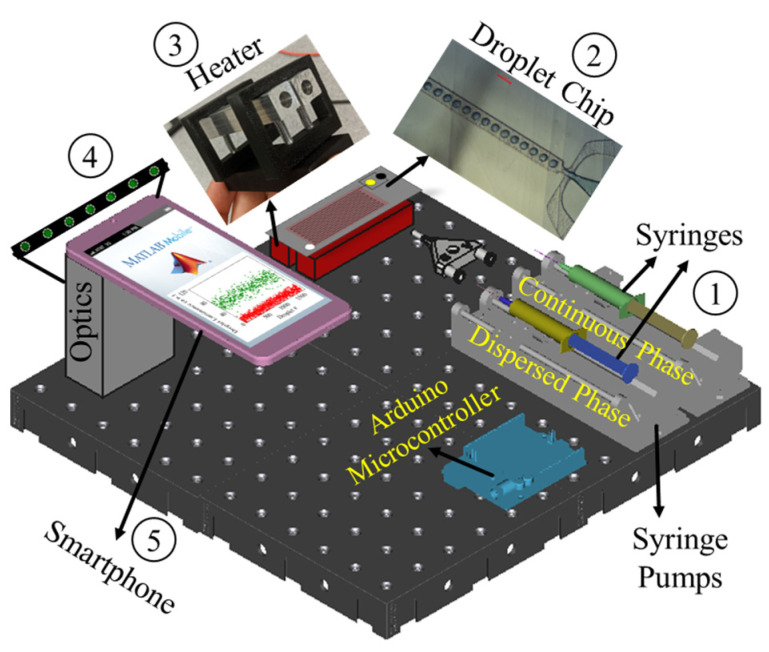
Schematic of the smartphone-enabled digital droplet LAMP (ddLAMP) platform. (1) Programmable syringe pumps drive the continuous phase (oil) and dispersed phase (LAMP assay mix) into (2) 3D printed microfluidic droplet generation chip. (3) An isothermal heater incubates the droplets flowing through the serpentine channels in the chip to actuate LAMP reactions. (4) The optical module interfaces the smartphone camera with a microscope and fluorescence filters to capture the flowing droplets. (5) An onboard image analysis program calculates the fluorescence intensity of each droplet to perform a quantitative LAMP assay.

**Figure 2 sensors-23-08310-f002:**
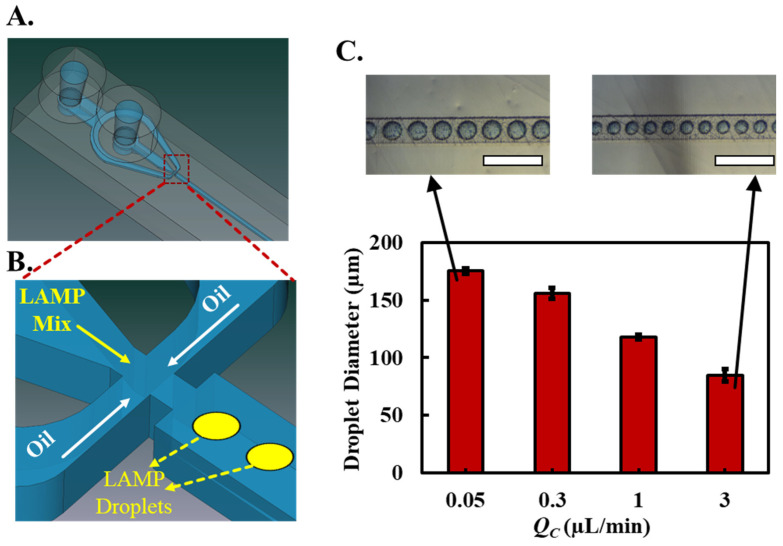
Fabrication and characterization of the microfluidic droplet generation chip. (**A**) Overview of the microfluidic chip with flow-focusing droplet generation segment. (**B**) Close-up of the flow-focusing segment depicting the use of oil as the continuous phase and LAMP assay mix as the dispersed phase. (**C**) Droplet size as a function of the continuous phase flow rate (*Q_c_*). Size variation droplet remains under ±5 microns for all synthesized droplets. Inset images depict the continuously flowing droplets corresponding to *Q_c_* = 0.05 and 3 μL/min (blue dye added to dispersed phase for droplet visualization; white scale bars represent 500 μm).

**Figure 3 sensors-23-08310-f003:**
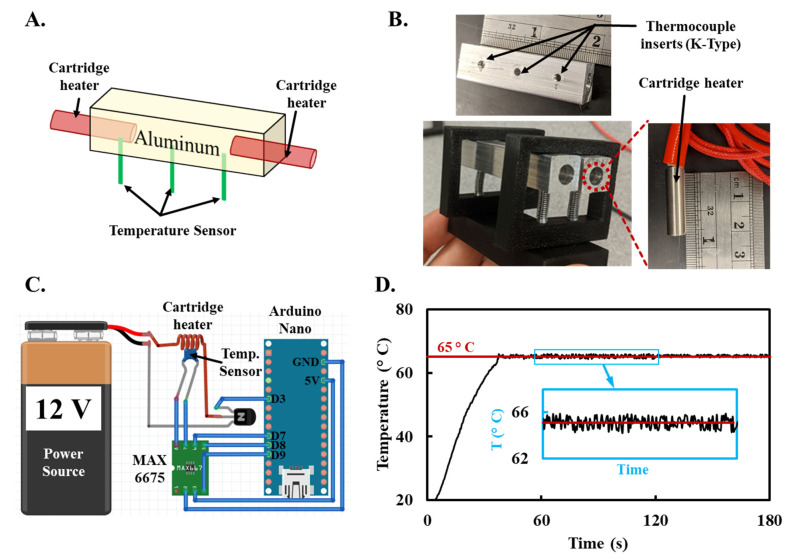
Engineering and performance evaluation of the low-powered heating blocks (**A**). Schematic representation of the machined aluminum heating blocks integrated with two heating cartridges (**B**). Assembled heating blocks comprising of spring-loaded machined aluminum blocks, cartridge heater, and holes to insert the thermocouple. (**C**). Circuit diagram detailing the Arduino-controlled heating module. (**D**). Performance assessment showing rapid attainment of the desired 65 °C setpoint, with deviations not exceeding ±0.8 °C from the setpoint.

**Figure 4 sensors-23-08310-f004:**
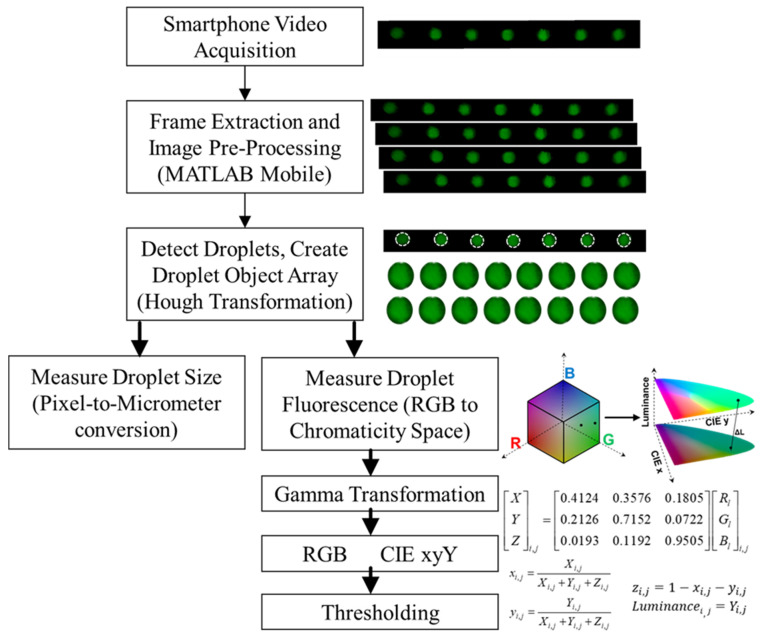
Smartphone-based droplet analysis using MATLAB Mobile The process flowchart starts with the video acquisition of droplets in a continuous flow through the field of view of the smartphone camera. Post-capture, individual frames are extracted using MATLAB Mobile to perform initial preprocessing of droplets such as Gaussian noise reduction and background subtraction. Droplets are detected using Hough transformation and stored in object arrays, detailing their RGB values, which are then utilized for size measurements and fluorescence intensity analysis. Droplet size determination is achieved via pixel-to-micrometer conversion. For fluorescence quantification, pixel values are transformed from RGB to CIE-xyY color space to determine the average luminance of each droplet.

**Figure 5 sensors-23-08310-f005:**
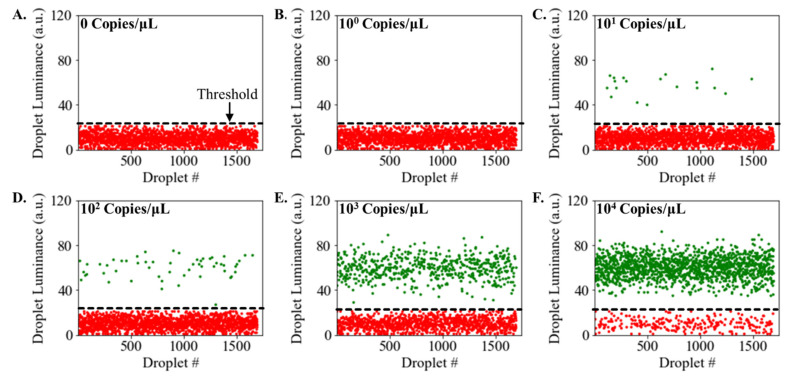
Detection of Positive and Negative Droplets using MATLAB Mobile. Detection of λ Phage DNA target using ddLAMP assay (**A**). Droplet luminance values from the negative control display dim fluorescence (**B**–**F**). Droplet luminance values from the samples containing target λ Phage DNA at concentrations ranging from 1 copy/μL to 10^4^ copies/μL. Green data points represent positive droplets (luminance > threshold), and red data points represent negative droplets (luminance < threshold value).

## Data Availability

Data can be provided upon request.

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
