# Peer review of "A Smartphone-Enabled Continuous Flow Digital Droplet LAMP Platform for High Throughput and Inexpensive Quantitative Detection of Nucleic Acid Targets"

_sensors, 2023, doi:10.3390/s23198310_

Round 1

Reviewer 1 Report

The manuscript suggested a digital droplet lamp platform. According to the manuscript, authors stressed the features as (1) 3D-printed microfluidic device, (2) heat control of aluminum-block, (3) smartphone-based fluorescence quantification, and (4) DNA concentration in each droplet with poison statistics. The contents will be paid attention to the readers of the journal Sensors. However, the following contents should be updated for the publication of the journal as,

[1] Throughout reading the manuscript, the reviewer did not find out features of the manuscript. Most of things which the authors stressed as features in the manuscript were already validated in the previous several journals. For example, why a 3D-printer microfluidic device was regarded as unique content? The others were also the same. If authors had different view unlike the reviewer, please show and stress it when compared with previous studies? For the reason, please write down precisely against features of the manuscript.

[2] With regard to order in manuscript contents, the manuscript wrote down as “IntroductionàResults àConclusionàMethods”. As a suggestion, how about changing the order of contents as “IntroductionàMaterials and methods àResults and discussion àConclusion”? There was no information on “Supplementary Materials”, which was noted in the manuscript. Please add (or upload) it.

[3] In the Figure 1, It was difficult to understand how to capture droplet image with your optical system. Please correct it.

[4] At last, the authors said the LAMP system was considered as portable and in-expensive. The reviewer also agreed that except two syringe pumps. Did it need an external electric power for operating syringe pump instead of 12 V battery?

Author Response

Please see the attached reviewer response sheet. 

Reviewer 2 Report

The article presents a low-cost method for quantitative detection of nucleic acid targets using smartphone and 3D printed microfluidics platform. The manuscript is very well written with good introduction and explanation of the platform. This low-cost method would have larger reach in low-middle income countries hence the article is of high value. 

The data, thought process, idea behind each component and its specifications are clearly written. Experimental design seems to be sound and the results correlate well with the hypothesis. 

The manuscript can be accepted with minor correction. Although the citations in the manuscript is very good, the last section of using CIE space and its analysis lack references. If the authors can add few references to this section, it would be easy for the readers to follow up.

Author Response

(The authors gave the same response as above.)

Reviewer 3 Report

Thee authors present an approach that integrates disposable 3D-printed droplet generator chips with consumer smartphone equipped with a custom image analysis application for conducting ddLAMP assays, thereby eliminating the necessity for expensive and complicated photolithographic techniques, optical microscopes, or flow cytometers. This interesting and valuable research work may be considered for publication, provided the authors address the reviewer comments mentioned below.

1. The introduction section is not comprehensive. It is important to highlight recent works in this domain.

2. The description related to Figure 4 needs to be improved.

3. How about the selectivity and specificity?

4. Authors present certain mathematical equations. Appropriate references should be provided. Else, the equations need to be derived.

5. Methods section needs to be presented much elaborately.

6. Interesting recent works in the domain of smartphone based and fluorescence based techniques such as surface plasmon-coupled emission (SPCE) can be referred to enhance the discussion section with better insights.

7. Relevant works in this domain may be included: PNAS, 2020, 117(37), 22727-22735. 

8. Did the authors perform any real time analysis of samples. Please comment.

Needs some improvement.

Author Response

(The authors gave the same response as above.)

Round 2

Reviewer 1 Report

Since the issues which the reviewer raised was discussed sufficiently in the revised manuscript, the reviewer suggested publication of the revised manuscript as current form. 

Author Response

This reviewer now suggests publication of the manuscript. 

Reviewer 3 Report

Authors address reviewer comments well.

Needs some improvement.

Author Response

This reviewer is satisfied with our revised manuscript.